# A Few TH-Immunoreactive Neurons Closely Appose DMX-Located Neuronal Somata Projecting to the Stomach Prepyloric Region in the Pig

**DOI:** 10.3390/ani10112008

**Published:** 2020-10-31

**Authors:** Jaroslaw Calka, Marta Ganko, Andrzej Rychlik

**Affiliations:** 1Department of Clinical Physiology, Faculty of Veterinary Medicine, University of Warmia and Mazury, Oczapowskiego 13, 10-718 Olsztyn, Poland; marta.ganko@uwm.edu.pl; 2Department of Clinical Diagnostics, Faculty of Veterinary Medicine, University of Warmia and Mazury, Oczapowskiego 14, 10-718 Olsztyn, Poland; rychlik@uwm.edu.pl

**Keywords:** dorsal motor vagal nucleus, stomach, prepyloric region, tyrosine hydroxylase

## Abstract

**Simple Summary:**

Although organization of the catecholaminergic system, in the porcine vagal motor nuclei of the pig, as well as distribution and chemical nature of the parasympathetic preganglionic neurons innervating the prepyloric region of the porcine stomach in the nucleus, have been well established, the question of a possible direct regulatory interaction between both neuronal systems still remains unknown. We discovered morphological foundations for direct regulatory action of the local TH-immunoreactive neurons on vagal preganglionic parasympathetic efferent neurons supplying the prepyloric region of the porcine stomach.

**Abstract:**

The vagus nerve is responsible for efferent innervation and functional control of stomach functions. The efferent fibers originate from neurons located in the dorsal motor nucleus of the vagus (DMX) and undergo functional control of the local neuroregulatory terminals. The aim of the present study was to examine the existence of morphological foundations for direct regulatory action of the local TH-immunoreactive neurons on parasympathetic efferent neurons supplying the prepyloric region of the porcine stomach. Combined injection of neuronal retrograde tracer Fast Blue into the stomach prepyloric region with TH immunostaining was used in order to visualize spatial relationship between DMX-located stomach prepyloric region supplying neuronal stomata and local TH-IR terminals. We confirmed existence of TH-immunoreactive neural terminals closely opposing the stomach prepyloric region innervating neurons at the porcine DMX area. The observed spatial relationship points out the possibility of indirect catecholaminergic control of the stomach function exerted through preganglionic parasympathetic efferent neurons in the pig.

## 1. Introduction

It is generally accepted that the dorsal motor vagal nucleus (DMX) and the nucleus ambiguous constitute a source of preganglionic parasympathetic innervation for the upper gastrointestinal tract through many species, including the cat [1], dog [2], rat [3], as well as the domestic pig [4,5]. The authors’ former tracing studies revealed that, in the pig, parasympathetic efferents supplying the stomach prepyloric region originated exclusively from the dorsal motor vagal nucleus [6], while the nucleus ambiguous innervated the esophagus [7].

The authors’ recent immunocytochemical study confirmed the cholinergic nature of the vagal preganglionic perikarya and additionally revealed that their direct surroundings were comprised of numerous networks of nerve terminals possibly affecting their regulatory functions [8]. Thus, between the cholinergic somata, numerous substance P (SP-), leu5-enkephalin (LENK-), cocaine- and amphetamine-regulated transcript (CART-), and nitric oxide synthase (NOS-IR) protrusions were found, while SP- and LENK-IR processes formed a basket-like structure closely surrounding parasympathetic cholinergic neurons.

Consistently, others have found catecholaminergic neurons in rat [9], dog [10], sheep [11], human [12], and swine [13] DMX. Although in the latter, fluorogold tracing studies demonstrated the presence of catecholaminergic neurons intermingled between parasympathetic efferent somata in the DMX area, the question of the spatial relationship between porcine parasympathetic efferent neurons and catecholaminergic neurons remains unclear. Consequently, the problem of the possible presence of morphological foundations for the direct action of catecholaminergic neurons on preganglionic parasympathetic DMX somata supplying the stomach in the pig remains uncertain. The question of the spatial correlation has recently gained special attention, since DMX α_2_-adrenergic receptors are likely to mediate central gastroprotective action in the stomach [14]. Thus, based on the gathered data, it is hypothesized that, in the porcine DMX, there could be a spatial correlation between “gastric” parasympathetic somata and local adrenergic neurons.

Therefore, the main goal of this project was to study the possible existence of direct contact or direct opposition between the DMX-located catecholaminergic neurons and the retrogradely-traced DMX originating stomach prepyloric region innervating neurons in the pig. Moreover, possible co-expression of TH in those vagal “gastric” neurons was also investigated.

## 2. Materials and Methods

All procedures were conducted on five (n = 5) immature gilts (about 20 kg of body weight) of the Large White Polish breed kept in standard laboratory conditions with access to species-specific feed (Grower, Nutrena, Poland, metabolic energy 12.9 MJ) and tap water. All experimental procedures were performed in agreement with the rules approved by the Local Ethics Committee in Olsztyn (decision no. 05/2010).

Before administration (15 min) of the main anesthetic, sodium thiopental (Thiopental, Sandoz, Kundl-Rakusko, Austria) (10 mg/kg of body weight i.v.), the animals were pretreated with azaperone (Stresnil, Jansen Pharmaceutica N.V., Belgium) (4 mg/1 kg of bodyweight, i.m.). The gilts were laparotomized to expose the stomach and a total volume of 50 µL of 5% aqueous suspension of the retrograde neuronal tracer Fast Blue (FB, EMS-CHEMIE GmbH, Germany) was multiple injected (n = 50, 1 µL each) with a Hamilton syringe into the diamond-shaped part (ca. 4 × 4 cm, ±16 cm^2^) of the diaphragmatic prepyloric area of the stomach (muscular and submucous layer) situated about 1 cm from the greater curvature of the stomach and 3 cm from the pylorus. To prevent leaking of the tracer outside the studied region, the needle was left in place for about 20 s after each injection.

Four weeks after FB injection, the gilts were euthanized by an overdose of sodium thiopental and transcardially perfused, with 4% buffered paraformaldehyde (pH 7.4).

Following perfusion, medulla oblongata blocks were collected from all animals, postfixed by immersion in the same fixative for 20 min, rinsed with 0.1 M PB (pH 7.4) over three days, and then transferred to 30% buffered sucrose solution (pH 7.4), containing 0.01% natrium azide, and stored at 4 °C. Finally, 14 μm thick cryostat sections were prepared and mounted on chrome alum-coated slides (about 1000 sections from each medulla were cut in total, about 300 of them contained FB-labeled “gastric” neurons and were further stained for TH immunoreactivity), and then analyzed under an Olympus BX51 fluorescent microscope (Olympus, Tokyo, Japan) equipped with an appropriate filter for FB to localize the FB-labeled perikarya. To avoid double counting the same neuron, only the FB-labeled cell bodies with a visible nucleus in every fourth section were scored.

Selected sections of the medulla containing FB-positive perikarya were processed for double-labeling immunofluorescence staining. Briefly, after air-drying at room temperature for 45 min and washing in a 0.1 M phosphate-buffered saline (PBS, pH 7.4, 3 × 10 min, the sections were incubated for 1 h, in a blocking buffer containing 0.1% BSA (bovine serum albumin) in 0.1 M PBS, 1% Triton X-100, 0.05% Thimerosal, 0.01% sodium azide, and again rinsed in PBS (3 × 10 min). Then, the sections were incubated overnight at room temperature with antisera (see Table 1), a mixture of ChAT (goat, AB144P-1ML, Millipore, Temecula, CA, USA, working dilution 1:50) and TH (mouse, MAB318, Millipore, Temecula, CA, USA, working dilution 1:200). On the following day, the sections were washed (PBS, 3 × 10 min) and incubated at room temperature, for 1 h, with secondary antibodies, i.e., Alexa Fluor 488 nm anti-goat (A11055, Thermo Fisher Scientific, Waltham, MA, USA, working dilution 1:1000) and Alexa Fluor 546 nm anti-mouse (A10036, Thermo Fisher Scientific, Waltham, MA, USA working dilution 1:1000). After consequent rinsing in PBS (3 × 10 min), the slides were coverslipped with carbonate-buffered glycerol (pH 8.6).

The omission of primary antisera and their replacement by normal sera were applied to control the specificity of immunofluorescence. Staining was not observed in either case.

Finally, the slides were analyzed and photographed under an Olympus BX51 microscope equipped with epi-fluorescence and appropriate filter sets.

## 3. Results

The stomach projecting prikarya, labeled with FB, were found to be mostly oval or round in shape. Multipolar forms were occasionally encountered. Their nuclei were situated centrally and somata were measured as 20 to 50 µm in diameter. They never co-expressed TH immunoreactivity, although all of them were ChAT-immunoreactive (Figure 1(1a–1c)).

Remarkably, among numerous FB-negative/ChAT-positive vagal perikarya, singular neurons were observed to express TH (Figure 1(1a–1c)). However, the remaining majority of the TH-IR neurons did not co-localize with ChAT. Catecholaminergic (FB-negative/ChAT-negative) perikarya were observed in the peripheral subregion of the DMX. The morphology of the majority of the TH-IR neurons varied from those of FB/ChAT-labeled cells. They were smaller in size (20 to 25 µm, occasionally 30 µm in diameter) and often possessed fusiform cell bodies with long processes and unstained nuclei.

Most interestingly, very close direct apposition of those TH-IR cells with FB-positive/ChAT-labeled “prepyloric” neurons was encountered. Quantification revealed that 0.329% of the “prepyloric” neurons established apposition with TH-IR cells. They occurred peripherally in the dorsal and ventral subregion. The processes of the catecholaminergic somata wrapped around the FB-positive/ChAT-positive neuronal cell bodies, thus, establishing a very close spatial relationship between the opposing neurons (Figure 1(2a–2c)).

## 4. Discussion

Combining injection of neuronal retrograde tracer Fast Blue into the stomach prepyloric region with TH immunostaining, the possibility of a close spatial relationship between vagal “prepyloric” motoneurons and TH-IR neurons in the porcine DMX region was demonstrated. Although Chaillou et al. [13] observed TH-IR neuronal, as well as vagal motor neuronal somata, in the porcine DMX, due to applied tracing procedure (transection of the left cervical vagal trunk), the authors were not able to specify the precise organ destination of the particular studied motoneuron. The selected approach allowed precise detection of the gastric “prepyloric” vagal motoneurons and their close spatial relationship with neighboring TH-immunoreactive neurons. The small number (see Table 2) of FB-labeled neurons apposing TH-IR terminals, found in the studied DMX, correlates with the limited surface area (±16 cm^2^) of the stomach wall under study. Consequently, evidence was provided that in the porcine DMX, gastric “prepyloric” neurons most likely underwent direct functional control of the local catecholaminergic system. Moreover, we delivered evidence that those gastric “prepyloric” cholinergic motoneurons did not co-express TH. However, since individual cholinergic motoneurons of an unknown destination (FB-negative) have been found to express TH immunoreactivity, one could expect that, in the pig, as previously described by Tsukamoto et al. [15] in the rat, some abdominal organs, including other not studied stomach areas as well as thoracic organs, may receive DMX-originating singular cholinergic/catecholaminergic vagal efferents. Elucidation of this question needs additional tracing investigations.

The current results, showing a close spatial relationship in the pig between vagal “prepyloric” somata and local TH-IR terminals in connection with a lack of TH-IR preganglionic neurons projecting directly to the prepyloric stomach region, corresponds with the results of Guo et al. [16] who found that, in the rat DMX, only 2% of the antrum projecting neurons expressed TH-IR. This interspecies similarity indicated that vagal DMX-located TH-IR neurons could indirectly, through preganglionic parasympathetic efferents, control stomach prepyloric functions. This hypothesis finds strong support in the results of Rosin et al. [17] and Tavares et al. [18], who reported abundant expression of α2A-, α2B-, and α2C-adrenergic receptor subtypes in neurons of the dorsal motor complex.

The presented results imply direct regulatory influence of the local DMX-located catecholaminergic neurons on the gastric vagal “preganglionic” somata. In the pig, the somata are also likely to be affected by other neuronal networks. Indeed, our former study [8] revealed networks of the SP-, LENK-, NOS and CART-immunoreactive processes located in close vicinity of the parasympathetic neurons. Nitrergic, as well as CART-IR terminals, formed basket-like structures closely surrounding DMX cholinergic neurons. Confirmation of such a functional relationship should come from the localization of the appropriate receptors on the vagal motoneurons.

The discovery of adjacent localization of the gastric vagal somata, in the current study, with nerve terminals originating from local DMX catecholaminergic neurons may constitute significant proof of the downregulatory effect of the vagal catecholaminergic neurons on the function of the vagal “prepyloric” parasympathetic projections. Nonetheless, further elucidation of this question, especially in the context of chemical adaptation of the neurons to catecholaminergic stimulation, requires additional studies.

## 5. Conclusions

The results of the current study may implicate direct regulatory action of the local catecholaminergic neurons on the stomach supplying vagal parasympathetic nerve cells in the dorsal motor vagal nucleus of the pig.

## Figures and Tables

**Figure 1 animals-10-02008-f001:**
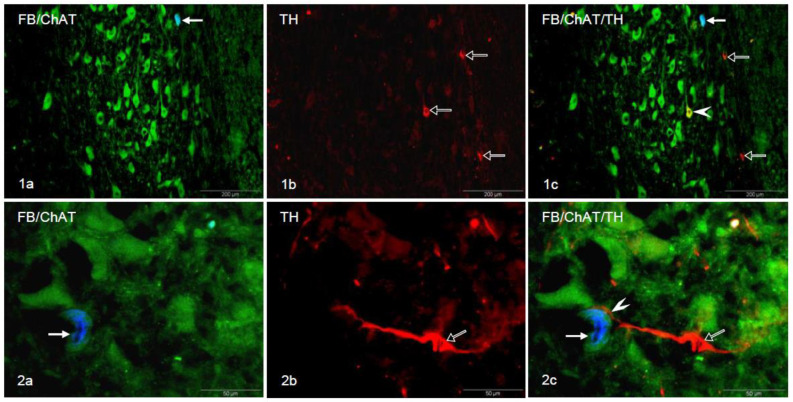
(**1a**) Double-labeled FB^+^/ChAT^+^ neuron (bold arrow) in the porcine DMX; (**1b**) TH^+^ neurons (empty arrows); (**1c**) Digital superimposition of pictures in (**1a**,**1b**). Double-labeled FB^−^/ChAT^+^/TH^+^ perikaryon (arrowhead), double-labeled FB^+^/ChAT^+^ neuron (bold arrow) and TH^+^ neurons (empty arrows) in the DMX. (**2a**) Double-labeled FB^+^/ChAT^+^ perikaryon (arrow) in the porcine DMX; (**2b**) TH^+^ neuron (empty arrow); (**2c**) Digital superimposition of pictures in (**2a**,**2b**). TH^+^ neuron (empty arrow) in close proximity of the FB^+^/ChAT^+^ perikaryon (arrow). The arrowhead indicates the place of direct opposition of the TH^+^ process on the FB^+^/ChAT^+^ neuronal soma.

**Table 1 animals-10-02008-t001:** Description of antibodies.

Antigen	Species	Dilution	Code	Manufacturer/Supplier
**Primary Antibodies**				
**TH**	Mouse	1:200	MAB318	Millipore, Temecula, CA, USA
**ChAT**	Goat	1:50	AB144P-1ML	Millipore, Temecula, CA, USA
**Secondary Antibodies**				
**Alexa Fluor 488 nm**	anti-goat	1:1000	A11055	Thermo Fisher Scientific, Waltham, MA, USA
**Alexa Fluor 546 nm**	anti-mouse	1:1000	A10036	Thermo Fisher Scientific, Waltham, MA, USA

**Table 2 animals-10-02008-t002:** Number of spatial appositions between FB^+^/ChAT^+^ and TH^+^ neurons.

Pig	I	II	III	IV	V	Mean ± Standard Error of Mean (SEM)
Number of appositions between CHAT^+^/FB^+^ and TH^+^ neurons	1	2	0	2	3	**1.6** **±** **0.5**
Number of FB^+^ cells	508	633	375	463	447	**485.2** **± 42.7**

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
