# Peer review of "A Few TH-Immunoreactive Neurons Closely Appose DMX-Located Neuronal Somata Projecting to the Stomach Prepyloric Region in the Pig"

_animals, 2020, doi:10.3390/ani10112008_

Round 1

Reviewer 1 Report

The authors have milden the conclusions from this preliminary study, and included some more of the requested information.

Reviewer 2 Report

This revised manuscript contains sufficient description of section analysis and a more cautious discussion of the results' significance. I have no further criticism. 

Reviewer 3 Report

The authors of this paper have address my previous concerns in their revision. I have no further comments.

This manuscript is a resubmission of an earlier submission. The following is a list of the peer review reports and author responses from that submission.

Round 1

Reviewer 1 Report

The manuscript by Calka et al investigate the proximity of TH+ projections and cholinergic neurons projecting to the stomach prepyloric region in the pig. Parasympathetic neurons of the dorsal nucleus were labelled retrogradely using Fast Blue, and tissue stained for TH and ChAT. Out of 2426 traced neurons, a handful where found to be in contact with TH+ axons. In sum, 0,03% of the neurons appeared to be contacted by TH+ axons. It is questinable whether these observed contacts have biological significance, or rather that these TH-axons may only be passing by. Given that the study was made on sections, the full length of axons were not possible to trace. It would have been appropriate to study these interactions in thicker sections (100-200 micrometer) which would have given a better confidence for the claimed interactions. Moreover, the size of the neurons are mentioned without any further statistics presented. Also for this aspect, thicker sections would be necessary. 

Author Response

Answer 1 Reviewer

The authors would like to thank you the reviewer for valuable critical comment.

To the Result section the sentence precising proportion of TH-IR appositions to the number of FB+/ChAT-IR neurons has been added.

“Quantification revealed that 0.329% of the “prepyloric” neurons established apposition with TH-IR cells.”

This is truth, found in our study proportion of  appositions 0.329% is not very high. However former study in the rat

  • L. Rosin, E. M. Talley, A.   Leeet al. Distribution of alpha-2C –Adrenergic Receptor-Like Immunoreactivity in the Rat Central Nervous System. J. Comp. Neurol. 372:135-165  (1996);
  • Agostinho Tavares, Diane E. Handy, Natalia N. Bogdanova, Douglas L. Rosene, and Haralambos Gavras,   Localization of α2A- and α2B-Adrenergic Receptor Subtypes in Brain.  1996;27:449–455).

Originally published1 Mar 1996https://doi.org/10.1161/01.HYP.27.3.449Hypertension. 1996;27:449–455).

report high expression of α2A- and α2B- and α2C- adrenergic receptor subtypes in the rat dorsal motor nucleus vagal neurons. Well, interspecies differences between rat and pig are possible, nevertheless this is tempting indication that regulatory significance of the local adrenergic system may be underestimated. Strong support comes also from observation that DMC vagal adrenergic stimulation exerts protective effect on the stomach mucosa. (Gyires, K.; Toth, V.E.; Zadori, Z.S. Gastric mucosal protection: from the periphery to the central nervous system. J. Physiol. Pharmacol. 2015, 66(3), 319-29.) We would like to deepen this problem in the pig DMC in our future work. Thank you for your suggestion concerning thickness of sections 100-200µm.

Reviewer 2 Report

This anatomical study intends to support the idea that TH-positive neurons in the dorsal vagal motor nucleus modulate cholinergic preganglionic vagal neurons innervating the pre-pyloric region. One example of a TH-positive process approaching a ChAT-positive cell body is documented. Although this should be a brief report, there are some problems to be addressed.

Major: 

1) It is not clear if this TH-positive process is a dendrite or an axon. Imaging should be improved by using confocal microscopy. This technique would allow not only for assessing the closeness of the relationship but also for a tentative distinction: axons are typically varicose while dendrites are typically smooth-contoured. All speculations about the possible significance of such contacts hinge on this data.

2) Even in a short communication, more detailed anatomical description is needed. For example, on which "peripheral subregion" of the DMX were the TH-positive neurons located? Dorsal? Ventral?

3) Likewise, a rough quantification should be done. Percentage of TH neurons versus ChAT positive neurons? How many "close relationships) of TH processes to ChAT perikarya?

4) Even if this additional data can be provided, the Discussion should be significantly shortened. At present, there is insufficient basis for functional speculations.

Author Response

Answer 2 Reviewer

The authors would like to thank you the reviewer for valuable critical comment.

  • This is truth apposition of TH-IR axons with studied “gastric” neurons is crucial for possible functional significance of these connections. However, supporting evidence for axonal character of these TH-IR processes come from study of Rosin et al. 1996 and Tavares et al. 1996.
  1. L. Rosin, E. M. Talley, A. Leeet al. Distribution of alpha-2C –Adrenergic Receptor-Like Immunoreactivity in the Rat Central Nervous System. J. Comp. Neurol. 372:135-165 (1996);
  2.  
  3. Tavares, D. E. Handy, N. N. Bogdanova, D. L. Rosene, and H. Gavras, Localization of α2A- and α2B-Adrenergic Receptor Subtypes in Brain Originally published1 Mar 1996https://doi.org/10.1161/01.HYP.27.3.449, Hypertension. 1996;27:449–455).

The researchers found in the rat dorsal motor nucleus vagal neurons plenty of  α2A- and α2B- and α2C- adrenergic receptor subtypes. Additionally, presence of  adrenergic receptors on vagal neurons combined with observation that DMC vagal adrenergic stimulation exerts protective effect on the stomach mucosa. (Gyires, K.; Toth, V.E.; Zadori, Z.S. Gastric mucosal protection: from the periphery to the central nervous system. J. Physiol. Pharmacol. 2015, 66(3), 319-29.) strongly support axonal character  of apposing TH-IR protrusions. We would like to study this interesting morphological relationship in the pig in our further study. Currently we run out of tissue sections from porcine DMC region since the project has been accomplished with publication of three other papers (Ganko M., Rychlik A., Calka J. 2013; Ganko M., Calka J. 2014; Ganko M., Calka J. 2014) and all suitable sections (containing FB+/AChE-IR have been utilized.

  • and 3) According to reviewer suggestions following supplementary sentences have been added to Result section: Quantification revealed that 0.329% of the “prepyloric” neurons established apposition with TH-IR cells. They occurred peripherally in the dorsal and ventral subregion. 

  • The Discussion section has been shortened (2 paragraphs have been removed) and modified with adding above mentioned papers of Rosin D. L. et al 1996 and Tavares A. et al. 1996. Generally the conclusion on direct contacts of apposing morphological structures has been soften in the text of manuscript.

Reviewer 3 Report

Introduction

The introduction provides a sufficient background including the gap and aims of the current study. However, a hypothesis is missing and should be added.

There are also several abbreviations not defined: SP, LENK, CART, NOS, and IR.

Methods

The methods is okay but could be improved via the suggestions below:

  1. What sex were the gilts?
  2. What was the species specific chow? Include the brand, place, and kJ content if possible.
  3. You mention you injected the tracer into the prepyloric area but did not mention what layer of the stomach you injected into. This is important as different layers may have different innervations.
  4. How many sections were analysed per animal?

Results

In table 2 you use commas instead of periods to denote decimal points. This makes the number difficult to read, please fix with periods.

Discussion

I have no issue with the discussion or conclusions.

Overall

The methods and conclusions from the authors are appropriate. In addition to the above comments I would suggest editing of the English language.

Author Response

Answer 3 Reviewer

The authors would like to thank you the reviewer for valuable critical comment.

Introduction

In the Introduction section missing hypothesis has been supplemented:

“Thus based on gathered data we hypothesize that in the porcine DMX might exist spatial correlation between “gastric” parasympathetic somata and local adrenergic neurons.”

Full names of SP, LENK, CART and NOS has been added:

“Thus, between the cholinergic somata numerous substance P (SP-), leu5-enkephalin (LENK-), cocaine- and amphetamine-regulated transcript (CART-) and nitric oxide synthase (NOS-IR) protrusions were found, ….”

Methods

  1. Gilts are young female pigs.

  1. Specificity of the chow has been added in the first sentence of the Materials section:

“All procedures were conducted on five (n=5) immature gilts (about 20 kg of body weight) of the Large White Polish breed kept in standard laboratory conditions with access to species-specific chow (Grower, Nutrena, Poland, metabolic energy 12.9MJ) and tap water.”

  1. The tracer (Fast Blue) was injected into muscular and submucous layers. Supplementation of the Methods section has been made :

“Fast Blue (FB, EMS-CHEMIE GmbH, Germany) was multiple injected (n=50, 1µl each) with a Hamilton syringe into the diamond-shaped part  (ca. 4cm x 4cm; ± 16 cm2) of diaphragmatic prepyloric area of the stomach (muscular and submucous layer) situated about 1 cm from the greater curvature of the stomach and 3 cm from the pylorus.”

  1. Concerning total number of cut and further processed sections supplementary statement has been added in Methods section:

“Finally, 14 μm thick cryostat sections were prepared and mounted on chrome alum-coated slides (about 1000 sections from each medulla were cut in total, approximately 300 out of them contained FB-labeled “gastric” neurons and were further stained for TH immunoreactivity) and then analyzed under a fluorescent microscope Olympus BX51 equipped with an appropriate filter for FB in order to localize the FB-labeled perikarya.”

Results

The commas in Table 2 have been changed on periods.

According to suggestion of the Referee the text of manuscript has been corrected by qualified translator.
